# Antibiofilm and Immune-Modulatory Activity of Cannabidiol and Cannabigerol in Oral Environments—In Vitro Study

**DOI:** 10.3390/antibiotics13040342

**Published:** 2024-04-09

**Authors:** Hernan Santiago Garzón, Manuela Loaiza-Oliva, María Cecilia Martínez-Pabón, Jenniffer Puerta-Suárez, Mayra Alexandra Téllez Corral, Bruno Bueno-Silva, Daniel R. Suárez, David Díaz-Báez, Lina J. Suárez

**Affiliations:** 1Programa de Doctorado en Ingeniería, Facultad de Ingeniería, Pontificia Universidad Javeriana, Bogotá 110231, Colombia; garzonhernan@javeriana.edu.co (H.S.G.); d-suarez@javeriana.edu.co (D.R.S.); 2Laboratory of Oral Microbiology, Faculty of Dentistry, University of Antioquia, Medellín 050010, Colombia; manuela.loaiza@udea.edu.co (M.L.-O.); mcecilia.martinez@udea.edu.co (M.C.M.-P.); jenniffer.puerta@udea.edu.co (J.P.-S.); 3Grupo Reproducción, Departamento de Obstetricia y Ginecología, Facultad de Medicina, Universidad de Antioquia, Medellín 050012, Colombia; 4Centro de Investigaciones Odontológicas, Facultad de Odontología, Pontificia Universidad Javeriana, Bogotá 110231, Colombia; tellezm@javeriana.edu.co; 5Department of Periodontology, Dental Research Division, Guarulhos University, Guarulhos 07023-070, Brazil; brunobue@fop.unicamp.br; 6Departamento de Biociências, Faculdade de Odontologia de Piracicaba, Universidade de Campinas (UNICAMP), Piracicaba 13414-903, Brazil; 7Unit of Basic Oral Investigation-UIBO, Facultad de Odontología, Universidad El Bosque, Bogotá 11001, Colombia; dadiazb@unbosque.edu.co; 8Centro de Investigaciones Odontológicas, Departamento del Sistema Periodontal, Facultad de Odontología, Pontificia Universidad Javeriana, Bogotá 110231, Colombia; 9Departamento de Ciencias Básicas y Medicina Oral, Facultad de Odontología, Universidad Nacional de Colombia, Bogotá 111321, Colombia

**Keywords:** cannabinoids, periodontitis, antimicrobial agents, immunomodulation, oral biofilms, cannabidiol, cannabigerol

## Abstract

Objective: To evaluate the in vitro antimicrobial and antibiofilm properties and the immune modulatory activity of cannabidiol (CBD) and cannabigerol (CBG) on oral bacteria and periodontal ligament fibroblasts (PLF). Methods: Cytotoxicity was assessed by propidium iodide flow cytometry on fibroblasts derived from the periodontal ligament. The minimum inhibitory concentration (MIC) of CBD and CBG for *S. mutans* and *C. albicans* and the metabolic activity of a subgingival 33-species biofilm under CBD and CBG treatments were determined. The Quantification of cytokines was performed using the LEGENDplex kit (BioLegend, Ref 740930, San Diego, CA, USA). Results: CBD-treated cell viability was greater than 95%, and for CBG, it was higher than 88%. MIC for *S. mutans* with CBD was 20 µM, and 10 µM for CBG. For *C. albicans*, no inhibitory effect was observed. Multispecies biofilm metabolic activity was reduced by 50.38% with CBD at 125 µg/mL (*p* = 0.03) and 39.9% with CBG at 62 µg/mL (*p* = 0.023). CBD exposure at 500 µg/mL reduced the metabolic activity of the formed biofilm by 15.41%, but CBG did not have an effect. CBG at 10 µM caused considerable production of anti-inflammatory mediators such as TGF-β and IL-4 at 12 h. CBD at 10 µM to 20 µM produced the highest amount of IFN-γ. Conclusion: Both CBG and CBD inhibit *S. mutans*; they also moderately lower the metabolic activity of multispecies biofilms that form; however, CBD had an effect on biofilms that had already developed. This, together with the production of anti-inflammatory mediators and the maintenance of the viability of mammalian cells from the oral cavity, make these substances promising for clinical use and should be taken into account for future studies.

## 1. Introduction

The use of *Cannabis sativa* (CS) and its derivatives has unequivocally increased over time due to the undeniable therapeutic benefits they offer. The remarkable medicinal properties of these substances have been extensively studied and well-documented, providing ample evidence of their efficacy in treating various medical conditions like Alzheimer’s, Parkinson’s, chronic and neuropathic pain, anxiety, schizophrenia, and inflammatory bowel disease, among others [1,2]. *Cannabis sativa* contains aromatic hydrocarbons known as cannabinoids and terpenes, which are found in the trichome cavity. Over 150 terpenes and 100 cannabinoids have been identified [3]. One of the most common phytocannabinoids is ∆9-tetrahydrocannabinol (∆9-THC), which produces psychoactive effects. Other cannabinoids include cannabidiol (CBD) and cannabigerol (CBG) [4]. Phytocannabinoids such as CBD and CBG are non-psychoactive and have been found to possess significant pharmacological activity, making them potentially useful for regular therapeutic use [5].

Therapeutic approaches with substances with concomitant antimicrobial and anti-inflammatory properties are widely appreciated today since the spectrum of inflammatory diseases initiated by infectious components is increasingly broader, and dysbiotic processes are becoming more clearly associated with inflammatory conditions every day [6,7], including oral diseases such as periodontitis [8,9]. The above, together with the great concern worldwide about the excessive use of antibiotics that has led to unprecedented levels of bacterial resistance [10,11], highlights the need to address these pathologies from both avenues of the problem: inflammation and antimicrobial strategies without the harming secondary effects of the antibiotics.

Additionally, there is the fact of the difficulty in treating biofilm-related diseases [12,13] (one of the main causes of deaths at the hospital level), given the characteristics of these complex structures that mean that the bacterial communities embedded in polymeric matrices are protected against the action of antimicrobials [14,15]. Bearing this in mind, there is a very important field of action for the use of Cannabis derivatives in the treatment of infectious/inflammatory diseases.

CBD, in particular, has been extensively researched and is known to be effective in reducing inflammation and acting as an antibiotic against various bacteria, including those that have developed resistance to other antibiotics: *Staphylococcus aureus*, *Enterococcus faecalis*, *Staphylococcus epidermidis*, *Pseudomonas aeruginosa*, and *Escherichia coli* [16,17]. Although research on CBG is limited, it has shown promise as an antibacterial agent, at least against *S. aureus* [18].

In addition to the antimicrobial effects, studies have also shown that CBD and CBG have a regulatory effect on the production of pro-inflammatory cytokines, while increasing anti-inflammatory ones, making them useful in managing conditions such as asthma, diabetes, and pancreatitis [19]. A reduction in the production of proinflammatory cytokines through pathways, such as the interruption of the NŦ-κβ pathway, has been demonstrated [18]. CBG has been shown to stop experimental intestinal inflammation and reduce the risk of developing colorectal cancer that is significantly increased in patients with ulcerative colitis [20].

There is currently a lack of information on the use of CBG and CBD in treating oral pathologies. However, studies have shown that CBG and CBD have antibacterial properties on oral microorganisms. In vitro models have demonstrated that CBG and CBD are effective against planktonic bacteria such as *Porphyromonas gingivalis* and *Treponema denticola* [21]. Moreover, they have been shown to be more effective in reducing bacterial colony-forming units compared with standard antiseptic products [22]. However, the evidence supporting CBD’s ability to avoid the formation and treat dental biofilms is weak.

In relation to interventions of the inflammatory process related to periodontitis, an experimental model of induced periodontitis in mice showed that the systemic administration of CBD led to positive effects on the host’s response, resulting in less bone loss, lower concentrations of RANK/RANKL, and smaller amounts of myeloperoxidase production [23]. In vitro and in vivo models have shown that both endo- and exo-cannabinoids are capable of decreasing the inflammatory response in periodontal tissue by blocking inflammatory pathways such as NF-κB, significantly reducing the production of inflammatory mediators and enhancing the release of IL-10 and TGF-β [24].

The utilization of oral Cannabis products in clinical settings is still limited, and further research is necessary to validate their widespread clinical application [22]. Furthermore, there is insufficient proof to demonstrate their effects on the structural cells responsible for protecting and supporting periodontal tissues. This study aimed to evaluate the in vitro antimicrobial and antibiofilm properties and the immune modulatory activity of CBD and CBG on cells and bacteria from oral microenvironments.

## 2. Results

### 2.1. Antimicrobial Activity of CBD and CBG on Planktonic S. mutans UA159 and C. albicans ATCC 10231

Both cannabinoids showed planktonic growth inhibition of *S. mutans* with a MIC of 20 µM and 10 µM for CBD and CBG, respectively. At the same time, for *C. albicans,* no effect was observed in any of the concentrations evaluated (MIC > 640 µM). Table 1 shows the MIC results for each compound and the activity controls.

### 2.2. Inhibition of Multispecies Biofilm Formation

In experimental design A, when the compounds were kept within the bacteria/biofilm from the beginning of the experiment, the ability of bacteria to form biofilms was inhibited by the presence of the compounds CBD and CBG. The metabolic activity of the subgingival biofilm was reduced by 50.38% with CBD at 125 µg/mL (*p* = 0.03) and 39.9% with CBG at 62 µg/mL (*p* = 0.023) before biofilm formation (Figure 1).

### 2.3. Metabolic Activity of Preformed Multispecies Biofilm

In experimental design B, when the 1 min daily treatments with compounds started on day 3, CBD exposure at 500 µg/mL reduced the metabolic activity of the previously formed biofilm by 15.41%. CBG did not have any effect (Figure 1).

### 2.4. Immunomodulatory Effect of CBD and CBG

The production of IL-2, CXCL10, IL-1β, TNF-α, and CCL2, IL-17 A, CXCL8, IL12p70, IL-6, IFN-γ, IL-10, TGF-β, and IL-4 by PLF after activation with CBD and CBG were measured by multiplex assay. For IL-2, CXCL10, IL-1β, TNF-α, and CCL2, the production was considered minimal (less than 5 pg/mL), regardless of the compounds’ concentrations and evaluation times. Even though the concentration recorded for IL-17 A, CXCL8, and IL12p70 was higher, none of them exceeded 20 pg/mL in any concentration or time. However, CBD significantly increased the production of IFN-γ in concentrations of 10 µM and 20 µM at 12 h of evaluation, (*p* = 0.0004, *p* = 0.0031, respectively).

For the regulatory cytokine, IL-10, production varied between 10 and 25 pg/mL, but none of the concentrations and times tested exceeded the control; nonetheless, with concentrations greater than 10 µM of CBD and CBG at 12 h, there was a tendency to increase the production, which was to be expected, being a late-producing cytokine. For TGF-β, as one of the main regulators of the immune response with powerful anti-inflammatory functions, CBG caused a significant increase at concentrations greater than 10 µM at 12 h exceeding 15 pg/mL (*p* = 0.012).

Similarly, the production of IL-4, a key regulator of human and adaptive immunity, was increased with stimulation with CBD at 1 µM in the three times evaluated (*p* = 0.0002). Notably, CBG in concentrations greater than 10 µM after 6 h of activation significantly increased IL-4 production compared with CBD activation levels (*p* = 0.0043). Thus, a considerable production of anti-inflammatory mediators such as TGF-β and IL-4, and a controlled production of IFN-γ, was induced by CBG at 6 and 12 h at a concentration of 10 µM compared with CBD at the same concentrations and times (Figure 2). (Appendix A).

### 2.5. Cytotoxicity Activity

Low cytotoxicity was observed in the culture of PLF in the presence of CBD and CBG. The cell viability of PLF was evaluated with the stimulation of CBD at different times and was higher than 95%, and for CBG, it was higher than 88%. There was a statistically significant difference between cell viability in the presence of CBG at 3 µM at 12 h, being lower compared with the control (*p* = 0.032). In addition, there was a statistically significant difference between the percentage of viable cells observed in the presence of CBD and CBG at 12 h, with the 3 µM concentration being higher for CBD (*p* = 0.023) (Figure 3).

According to the international standard guide (DIN EN ISO 10993-5:2009, German Institute for Standardization, Berlin, Germany) for the classification of cytotoxicity, neither of the two compounds presented cytotoxic effects at any of the times and concentrations evaluated since cell viability was always greater than 80%. According to this guide, compounds derived from plants are considered non-cytotoxic when cell viability is less than 25%, slightly toxic if the inhibition is between 25 and 50%, moderate between 50 and 75%, and high when it is greater than 75%, concerning the control group [25].

## 3. Materials and Methods

The research protocol was approved by the Ethics Committee of the Pontificia Universidad Javeriana (Act 005-2021).

### 3.1. Materials

CBD and CBG were obtained commercially with a purity of 99% characterized by high-performance liquid chromatography (Avicanna, Toronto, ON, Canada). Cannabinoid stock solutions were prepared in DMSO (Thermo Scientific™, Waltham, MA, USA), stored at 4 °C, and protected from light. Chlorhexidine (C9394, Sigma-Aldrich, St. Louis, MO, USA) was used as activity control for antimicrobial assays. The antifungals amphotericin B and fluconazole (Sigma-Aldrich, USA) were also used as positive controls to evaluate anti-*Candida albicans* activity.

Fibroblasts derived from the periodontal ligament (PLF) obtained from the Dental Research Center of the Faculty of Dentistry of the Pontificia Universidad Javeriana were used for the cytotoxicity assays. Cells were cultured according to ATCC recommendations with Dulbecco’s modified essential medium (D-MEM, Gibco^TM^, Paisley, UK), supplemented with 10% Fetal Bovine Serum (SFB, Sigma-Aldrich, USA), 100 U/mL penicillin, 100 ug/mL of streptomycin (Sigma-Aldrich, USA).

### 3.2. Microbiological Tests

#### 3.2.1. Antimicrobial Activity and CBD and CBG’s Minimum Inhibitory Concentration (MIC)

The determination of the antimicrobial activity of the cannabinoids CBD and CBG was carried out through microdilution assay, following the recommendations of CLSI M07 [26] and with minimal modifications according to the technical needs of the test. Resazurin was used as a visual revealer of bacterial growth [27].

From a new culture of *S. mutans*, 3 to 5 colonies were selected and transferred to a test tube with 3 mL of BHI broth (Oxoid, Basingstoke, UK); this was incubated at 37 °C, with 5% CO_2_ for 20 to 24 h. From the previous culture, the final inoculum of *S. mutans* was prepared at a concentration of 3–8 × 10^5^ CFU/mL, according to previous standardization. A 96-well plate was prepared with concentrations of 40 µM to 0.3125 µM of CBD and CBG in triplicate and chlorhexidine (Sigma-Aldrich, USA) as activity control at concentrations of 150 µg/mL to 1.17 µg/mL. Subsequently, each well was inoculated with 7 µL of the previously standardized microbial inoculum, except for the sterility control wells. The plate was incubated for 24 h at 37 °C in 5% CO_2_. After the incubation, 30 µL of 0.015% resazurin was added to each well and incubated for another 3 h. Then, the MIC was determined visually, defined as the lowest concentration of the compound that prevents the color change (from blue to pink/orange).

The biological activity of CBD and CBG compounds against *C. albicans* ATCC 10231 was evaluated by microdilution assay, following the recommendations of CLSI M27 (4th edition) with some modifications based on the technical needs of the assay. The minimum inhibitory concentration (MIC) was determined to assess the effectiveness of these compounds [28].

*C. albicans* ATCC 10231 was reactivated in aerobic conditions on Sabouraud dextrose agar at 37 °C. The inoculum suspension of *C. albicans* was prepared from new colonies after 24 h of incubation. The plate was prepared with serial dilutions from 640 µM to 5 µM for each cannabinoid. A total of 100 µL of each concentration (2X) were suspended in 96-well microplates. Subsequently, *C. albicans* inoculum (2X) was prepared by counting in a Neubauer chamber, and 100 µL of *C. albicans* blastoconidia at 2.5 × 10^3^ CFU/mL (2X) was added to each well to the challenge plate, except for sterility controls. The plate was incubated for 24 h in 5% CO_2_ at 37 °C. Three assays were performed in triplicate. The minimum inhibitory concentration (MIC) was determined to be the lowest concentration of CBD or CBG, which inhibited 90% of the visible growth of *C. albicans* relative to the growth control.

#### 3.2.2. CBD and CBG’s Antibiofilm Activity against Subgingival Multispecies Biofilm

The design of this part involved two laboratory experiments that aimed to reproduce the possible clinical indications of CBD or CBG. In the first experimental design (A), compounds were kept with the bacteria (biofilm) from the very beginning of the experiment. In the second experimental design (B), the bacteria were allowed to form an initial biofilm for 72 h. Then, treatments with both compounds separately were initiated, being performed twice a day, for 1 min for each treatment. Both experimental designs used CBD and CBG at 62.5, 125, 250, and 500 µg/mL. The positive control was 0.12% chlorhexidine, and the biofilms were treated with culture media as negative control.

##### Multispecies Biofilm Formation

In vitro multispecies biofilm cultures were prepared with 33 bacterial species, as described by de Figueiredo et al. 2019 [29], with minimal modifications. The strains used were *Actinomyces gerencseriae* (23860), *Actinomyces israelii* (12102), *Actinomyces naeslundii* (12104), *Actinomyces oris* (43146), *Actinomyces odontolyticus* (17929), *Veillonella párvula* (10790), *Streptococcus gordonii* (10558), *Streptococcus intermedius* (27335), *Streptococcus mitis* (49456), *Streptococcus oralis* 35037, *Streptococcus sanguinis* (10556), *Streptococcus anginosus* (33397), *Streptococcus mutans* (25175), *Aggregatibacter actinomycetemcomitans* (29523), *Capnocytophaga gingivalis* (33624), *Capnocytophaga ochracea* (33596), *Capnocytophaga sputigena* (33612), *Eikenella corrodens* (23834), *Campylobacter gracilis* (33236), *Campylobacter showae* (51146), *Eubacterium nodatum* (33099), *Eubacterium saburreum* (33271), *Fusobacterium nucleatum* subsp. *polymorphum* (10953), *Fusobacterium nucleatum* subsp. *vincentii* (49256), *Fusobacterium periodonticum* (33693), *Parvimonas micra* (33270), *Prevotella intermedia* (25611), *Streptococcus constellatus* (27823), *Treponema denticola*, *Tannerella forsythia* (43037), *Porphyromonas gingivalis* (33277), *Gemella morbillorum* (27824), *Propionibacterium acnes* (11827), *Selenomonas noxia* (43541).

Tryptone soy agar (Scharlau, Barcelona, Spain) with 5% sheep blood was used to grow most species under anaerobic conditions, 85% nitrogen, 10% carbon dioxide, and 5% hydrogen. *Porphyromonas gingivalis* was grown on tryptone soy agar containing yeast extract enriched with 1% hemin, 5% menadione, and 5% sheep blood. *Tannerella forsythia* was grown on tryptone soy agar containing yeast extract enriched with 1% hemin, 5% menadione, 5% sheep blood, and 1% N-acetylmuramic acid. All species were allowed to grow on agar plates for 24 h and then transferred to glass tubes containing Brain Heart Infusion (BHI) culture medium (Becton Dickinson, Sparks, MD, USA) supplemented with 1% hemin. After 24 h of growing on conical tubes, the optical density was adjusted for the inoculum to have about 10^8^ cells/mL of each species. A dilution of individual cell suspensions was performed, and 100 μL aliquots containing 10^6^ cells from each species were added to 11,700 μL of BHI broth complemented with 1% hemin and 5% sheep blood to obtain an inoculum of 15 mL [29].

The multispecies biofilm model was developed using a Calgary biofilm device in a 96-well plate (Nunc; Thermo Scientific, Roskilde, Denmark). A 150 μL aliquot of each inoculum was added to the wells, corresponding to ~1 × 10^4^ cells of each bacterial strain, except for *P. gingivalis* and *Prevotella intermedia*, whose inoculum was adjusted to 2 × 10^4^ cells. A lid containing polystyrene pins was used to seal the 96-well plate (Nunc TSP system; Thermo Scientific, Roskilde, Denmark). Coated plates were incubated at 37 °C under anaerobic conditions.

For experimental design A, the compounds were included on the first day of biofilm formation, together with the bacteria inoculum. On day three, the spent medium (BHI broth with 1% hemin and 5% sheep blood) was replaced for both experimental designs (A and B). For experimental design A, the compounds were included once more on day three and kept within the biofilm until collection. For experimental design B, on day 3, the two 1 min daily treatments were started and maintained until day 6, totalizing eight 1 min treatments. On day 7, the biofilms were collected for subsequent analysis.

##### Determination of Multispecies Biofilm Metabolic Activity

The effects of compounds and controls on the metabolic activity of multispecies biofilm cells were measured in a spectrophotometric assay with 2,3,5-triphenyltetrazolium chloride (TTC) (catalog No. 17779; Fluka analytical). TTC was used to differentiate between metabolically active and inactive cells. TTC white substrate was enzymatically reduced to red formazan by live cells due to the activity of several dehydrogenases. The change in substrate color was an indirect measure of bacterial metabolic activity.

The pins were transferred to 96-well plates with 200 μL/well of fresh BHI medium supplemented with 1% hemin and 0.1% TTC solution to measure biofilm cells’ metabolic activity. The plates were incubated under anaerobic conditions for 8 h at 37 °C. TTC reduction to red formazan was read at 485 nm in a spectrophotometer (Agilent BioTek EPOCH, Santa Clara, CA, USA) [29].

### 3.3. Cytotoxicity Assessment

The cytotoxicity of CBD and CBG was evaluated by flow cytometry with propidium iodide according to the protocol of Riccardi and Nicoletti (2016) [30] with minimal modifications. For all tests, CBD and CBG were used with a purity of 99.9%. CBD and CBG were reconstituted in dimethyl sulfoxide (DMSO, Thermo Scientific™) at a concentration of 0.01% and evaluated between 0.5 µM and 20 µM based on previous reports in the literature [31]. CBD and CBG compounds were tested at concentrations of 0.5, 1, 3, 10, and 20 µM at 3, 6, and 12 h. The positive control was 100 µM hydrogen peroxide, and the negative control was DMSO (0.01%).

A suspension of 1 × 10^6^ cells (PLF) in 1 mL PBS in 12 × 75 mm tubes (Biolife, Bothell, WA, USA) was centrifuged at 200× *g* for 5 min at room temperature, the PBS was removed, and the cell pellet was resuspended in 1 mL of fluorochrome solution (Becton Dickinson, Franklin Lakes, NJ, USA). Tubes were placed in the dark at 4 °C before flow cytometry for at least 1 h and no more than 24 h. The LSR Fortessa II cytometer (Becton Dickinson) with a 488 nm laser line was used for excitation. Red fluorescence (>600 nm) and dispersion were measured, collecting at least 20,000 events. The FlowJo^®^ 8.7 software (Tree Star, Inc., Ashland, OR, USA) was used to carry out the analysis.

### 3.4. Evaluation of Cytokine Production by Multiplex Assay Technique

Cytokine quantification was performed using the LEGENDplex kit (BioLegend, San Diego, CA, USA) Ref 740930. CBD and CBG at 0,5, 1, 3, 10, 20 uM concentrations were added to PLF cultures at 3, 6, and 12 h and the production of IL-2, CXCL10, IL-1β, TNF-α, and CCL2, IL-17 A, CXCL8, IL12p70, IL-6, IFN-γ, IL-10, TGF-β, and IL-4 were measured. The negative control was cell culture supernatant of periodontal ligament fibroblasts in base medium (DMEM) with DMSO at a concentration of 0.01%. The LSR Fortessa II cytometer (Becton Dickinson) was used, and the results were generated by the LENGENDplex Data Analysis Software v8.0 (BioLegend, USA). The protocol used was the one provided by the manufacturer. The cytometer reading was simultaneous for all conditions [32].

### 3.5. Statistic Analysis

The determination of biofilm metabolic activity results for CBD and CBG were analyzed using two different methods: (1) comparison of concentrations (µg/mL) between test groups before and after biofilm formation; (2) percentage of absorbance with respect to the viability mean of the negative control group before and after biofilm formation.

The data distribution was preliminarily evaluated for all analyses using the Shapiro–Wilk test, which indicated normal distribution (*p* > 0.05) in all comparison scenarios. Consequently, the mean and standard deviation were used as summary measures, and parametric tests were used to compare the groups.

Comparison of concentrations and viability percentages was performed using one-way ANOVA with Bonferroni post hoc test. The assumption of homoscedasticity was checked using the Bartlett test for equal variances. The comparison of the outcomes of interest between the presence or absence of biofilm in each treatment was carried out using the *t*-test. All analyses were performed using IBM’s SPSS Statistics V24 statistical package. The level of statistical significance was set at *p*-value ≤ 0.05.

The statistical analysis was performed using IBM’s SPSS Statistics V24 statistical package (Chicago, IL, USA) for the cytotoxicity and cytokine production tests. Comparisons were analyzed via one-way ANOVA, with the level of statistical significance set at *p*-value ≤ 0.05.

## 4. Discussion

The antimicrobial properties of CBD have been known since 1976 when van Klingeren and Ham published a report [33] stating that it had a MIC between 1 and 5 µg/mL^−1^ against *Staphylococcus* and *Streptococcus*. However, they found no activity against *E. coli*, *Salmonella* Typhi, or *Proteus vulgaris* (MIC > 100 µg/mL^−1^). Despite this potential, evidence for its antimicrobial properties were overlooked until 2008 when Appendino et al. (2008) published a compelling study demonstrating its efficacy against six strains of methylin-resistant *Staphylococcus aureus* (MARS), with a MIC between 0.5 and 2 µg/mL^−1^ [16].

Some studies show a MIC between 1 and 4 µg/mL^−1^ (3.17–12.7 µM) in a diverse range of gram-positive bacteria, including Methylin-resistant *Staphylococcus aureus*, *Streptococcus pneumoniae*, *Enterococcus faecalis*, and anaerobic *Clostridioides* (formerly Clostridium) *difficile* and *Cutibacterium* (formerly *Propionibacterium*) *acnes*. Interestingly, these MIC values did not change appreciably against highly resistant bacteria, such as *S. aureus*, vancomycin-resistant enterococci, and the hypervirulent *C. difficile* strain 027. However, CBD was less potent against some beta-hemolytic *Streptococci pyogenes* and *S. agalactiae* with MIC values between 8 and 32 µg/mL (25.4–101.7 µM). It was inactive against 20 species of gram-negative bacteria: *E. coli*, *Klebsiella pneumoniae*, *Pseudomonas aeruginosa*, and *Acinetobacter baumannii*. Surprisingly, it had excellent potential against four gram-negative bacteria: *Neisseria gonorrhoeae* (MIC 1–2 μg/mL^−1^), *Neisseria meningitidis* (MIC 0.25 μg/mL^−1^), *Moraxella catarrhalis* (MIC 1 μg/mL^−1^), and *Legionella pneumophila* (MIC 1 μg/mL^−1^). CBD was not effective against the efflux pump of some strains of *E. coli* or *P. aeruginosa* (MIC >128 μg/mL^−1^), thus attributing that gram-negative bacteria capable of generating this resistance mechanism would not be sensitive to CBD [34].

In the present investigation, *S. mutans*, a commensal facultative anaerobic gram-positive bacterium that is part of the biofilms of the oral cavity, was used in a planktonic state, on which a MIC of 20 µM was obtained for CBD. Feldman et al. (2020) showed that the effect on *S. mutans* of CS extracts is related to some endocannabinoid system components, such as N-arachidonoylethanolamine, oleoylethanolamide, palmitoylethanolamide, and stearoyl ethanolamide, and their mixtures with poly-L-lysine. The results showed that in the planktonic state, none of the components separately had a MIC value in all the doses tested from 0 to 25 µg/mL. However, the combination of N-arachidonoylethanolamine at 6.25 µg/mL, 12.5 µg/mL, and 25 µg/mL with poly-L-lysine 25 µg/mL reduced bacterial growth by 26%, 54%, and 71%, respectively. On the other hand, the mixture between oleoylethanolamide and poly-L-lysine in doses between 6.25 µg/mL and 25 µg/mL reduced the growth of the bacteria between 80% and 86% [35].

It is worth highlighting that CBD boasts a strong antibacterial impact, not only against planktonic bacteria, but also in inhibiting biofilm formation and halting the growth of both gram-positive and gram-negative bacteria that are impervious to existing antibiotics. The utilization of small therapeutic doses, coupled with the non-development of resistance, even with repeated use, make the application of these cannabinoids highly advantageous [34].

As for the antibacterial activity of CBG on oral bacteria, it has been shown that for *S. mutans* in the planktonic state, CBG exerts a bacteriostatic effect that is affected by the initial bacterial cell density, with a MIC of 2.5 µg/mL, and also an antibacterial effect against *S. sanguis*, *S. sobrinus*, and *S. salivarius*. CBG also causes alterations in the cell membrane through immediate hyperpolarization, increased permeability, and the accumulation of mesosome-like structures. As an additional effect, it prevents the acidification of the medium caused by *S. mutans*, suggesting an anti-cariogenic activity [36].

In the present work, the MIC for *S. mutans* in the planktonic state was 10 µM. CBG is one of the most potent cannabinoids against other gram-positives, such as *Staphylococcus aureus*. Compared with the traditional antibiotics, norfloxacin and erythromycin, it has a significantly lower MIC (1 µg/mL), with even better results than CBD. In addition, it also presented a lower MIC than tetracycline and oxacillin in at least one of the six strains [16]. It has also been reported that in a planktonic state, the MIC for *S. aureus* was 2 µg/mL (6.3 µM) using CBG, and as such was the extract with the highest anti-biofilm activity, since with a dose of 0.5 µg/mL, the formation of the biofilm was inhibited by 50%. However, a higher dose was required to affect the bacteria in the planktonic state. Additionally, CBG can eradicate pre-formed biofilms of USA300-resistant *S. aureus* at a concentration of 4 µg/mL (12.6 µM). Future studies should evaluate whether this moderate antibiofilm outcome would be effective in vivo. Typically, the first step in natural product bioprospecting is in vitro assays. Subsequently, the natural product-derived molecules should undergo in vivo studies, including animal models, and ultimately clinical trials. The vehicle, route of administration, and pharmacokinetic properties can all influence whether the positive effects observed in vitro will be reflected in vivo [37].

Cannabis derivatives’ bactericidal action mechanism is still under study; it is considered to be related to alterations in the bacterial cytoplasmic membrane. It has been found that CBG can be effective against gram-negative bacteria only when the outer membrane is previously permeabilized [38]. However, one of the conditions that should be considered for a possible systemic administration of CBG is its potent effect on the α-2 receptor, which can induce hypotension, bradycardia, and xerostomia. However, it is a drug still under study and must be tested in humans [18].

The first report on the fungicidal effect of ethanol and petroleum, ether extracts of cannabinoids, was undertaken in 1995 by Wasim et al., against *Candida albicans* and *Aspergillus niger* at 10 mg/mL and 5 mg/mL, respectively [39]. Nevertheless, the cause of this impact could not be pinpointed to specific elements. Ali et al. challenged these results in 2012, demonstrating that extracts of petroleum ether from a Sudanese plant exhibited antibacterial effects against gram-positive bacteria, yet showed no activity against fungi [40]. This suggests that the origin of the plant may eventually generate different effects, in addition to the fact that the extracts may not be very specific to attribute their action to one component or another.

Concerning the antifungal activity of CBD, studies are scarce. For CBG, no evidence has been reported on fungicidal activity in *C. albicans*. In the present work, at a concentration of 640 μM of CBD and CBG, there was no evidence of fungicidal activity against *C. albicans* in the planktonic state. The findings of the present study are lower than what was found by Feldman et al. in 2021, who used CBD (99.4% purity) and concentrations between 3.25 and 400 µg/mL (10.3–1271.9 µM) in *C. albicans* as a planktonic fungus. The MIC could not be established [26]. Even though there was no effect on the planktonic C. *albicans*, an essential contribution of the cited study was to show that it disrupted the biofilm formation with a dose-dependent effect. After 24 h of incubation, 12.5 µg/mL (39.7 µM) inhibited biofilm formation by 37%, while at 72 h, a concentration of 1.56 µg/mL (4.96 µM) was sufficient to inhibit 31% of biofilm formation. Additionally, with a low dose of 1.56 µg/mL (4.9 µM), mature biofilm was reduced by 28% compared with the control, and with a dose of 3.12 µg/mL (9.9 µM), mature biofilm formation was disrupted at 44%. The metabolic activity of the mature biofilm was reduced to 32–46% when treated with higher concentrations between 6.25 and 100 µg/mL (19.8–317 µM) of CBD [26].

In previous studies, other components of the Cannabis plant with antifungal activity have been reported: anandamide and arachidonyl serine against *C. albicans*, which prevents the adhesion of hyphae to epithelial cells and inhibits yeast–hyphal transition and hyphal growth, without affecting the viability of *C. albicans* [27].

To test the possibility of immune modulation with CBD and CBG on PLF, the Legendplex kit was used for the quantification of the extracellular cytokines present once the PLF were stimulated at the exact times and concentrations of the cell viability experiment. Thus, it was possible to quantify 13 cytokines in the supernatants: IL-4, IL-2, CXCL10, IL-1β, TNF-α, CCL2, IL-17A, IL-6, IL-10, IFN-γ, IL-12p70, CXCL8, TGF-β. These cytokines have different functions according to their nature, which can generate inflammatory or regulatory processes in the immune system. Therefore, a detailed analysis of each one or subgroup is necessary.

Throughout the experiment, the production of TNF-α and CCL2 remained consistently negligible, with the concentrations assessed for both compounds indicating that they do not exhibit a pro-inflammatory nature. In this context, while there is a lack of in vitro models utilizing cell lines derived from the periodontal ligament under this particular stimulus, the minimal production of TNF-α plays a crucial role as it indicates an anti-inflammatory effect of CBD and CBG. In reference to CBD, authors such as Sangiovanni et al. in 2019, in a model of skin keratinocytes and fibroblasts, found that with a stimulus for 6 h at a concentration of 2.85 µM, there was a significant reduction in TNF-α production, attributing this effect to the fact that the NŦ-κβ pathway was altered, as one of the main inflammatory pathways [28]. This has also been shown in in vitro models with neuronal cells [41]; furthermore, in an in vivo model, it was found that the inhibition of NŦ-κβ by CBD infusion improved cerebral ischemia in rats [42], in addition to increasing the activation of the transcription factor STAT3 and decreasing the activation of STAT1 induced by LPS in BV-2 microglial cells.

It has also been shown that with LPS induction of microglial cells in mice, CBD in concentrations between 1 and 10 µM potently inhibited the release of the cytokines TNF-α and IL-1β as primary mediators of inflammation [42]. The impact of CBG on neuroinflammation models induced by LPS also demonstrates its ability to enhance the reduction of cytokines like TNF-α, IL-1β, and IFN-y, as well as oxidative stress [43].

Although different immune regulation mechanisms attributed to the effect of CBD have been described, such as the differentiation of T lymphocytes into regulatory T lymphocytes and even Treg17 [44], in recent years, one of the main mechanisms attributed to the control of inflammation in both healthy and diseased human cells is the disruption in the NŦ-κβ pathway [45,46]. Although no cell signaling analysis was performed in the present investigation, the null production of TNF-α may suggest an interruption of this powerful inflammatory pathway, as a result of the action of the compound on this cell line, which would also be sensitive to this effect, even in short stimulation times and CBD and CBG being used at low concentrations. In a study by Chiricosta et al., a transcriptomic profile was made in gingival mesenchymal stem cells, which found that genes involved in TNF-α signaling, such as MAP3K7, CLIP3, and CASP8, are downregulated with CBD and moringa treatment, a finding that is also in agreement with what was found in this investigation [47].

The evidence about CBG, in general, is scarce. In a model of neuroinflammation with a CBG pretreatment, the production of IL-1β, TNF-α, and IFN-γ was reduced, and it is also attributed to the exact mechanism of intervention in the NŦ-κβ pathway and the MAPK pathway, in concentrations from 1 µM to 20 µM [43]. These two combined extracts of CBD and CBG have also been tested, at concentrations of 2.5 to 5 µM, showing a reduction in the NŦ-κβ pathway and increasing the production of IL-10 and IL-37 when CBG is added in a 5 µM dose. It decreases apoptosis by downregulating Bax proteins and upregulating Bcl-2 expression [48]. These findings would be consistent with what was found in the present investigation since the levels of these cytokines were low in the two compounds at the different times evaluated. With this, it can be suggested that at least these two cannabinoids eventually have this same pathway of action that can be enhanced by mixing them; however, more research is required in this regard.

The production of IL-2, IL-1β, and the chemokines CLC2 and CXCL10 did not exceed 5 pg/mL for the two compounds. This finding is also in agreement with Nichols et al. in 2020, who described that these cytokines are critical targets in the action of CBD in different cell lines such as microglial, endothelial, and hepatic; therefore, a low production would be expected, as also evidenced in the present investigation [45]. This is another possible way of controlling inflammation. However, its mechanism still needs to be precise.

The cytokines IL-17A and the precursor of IL-8, CXCL8, (despite their strongly pro-inflammatory nature) did not exhibit a significant increase beyond 10 pg/mL in any of the concentrations and time frames assessed for both compounds. This suggests that stimulating PLF with these compounds can result in the extracellular production of these pro-inflammatory cytokines, albeit to a minimal extent. Although there are no related studies on these types of cells, it has been described that IL-17 signaling is one of the critical pathways suppressed by CBD (5 µM) in vitro in T lymphocytes [31]. Although it has also been described that CBD can suppress the production of IL-6 and, therefore, the differentiation of Th17 cells. With a concentration of 3.2 to 64 µM, the production of IL-17 A was reduced in CD3+ T cells [49]. Studies linking IL-17 A and CBD are scarce, but an increase in Treg17 in cells treated with CBD has been reported [44].

The only report found regarding the action of CBG on the Th17 profile was in 2014 and is also related to a neuroinflammation model, where a CBG derivative (VCE-003) was tested, which had an inhibitory effect on the Th1 and Th17 profiles, therefore, caused the interruption of IL-17 production [50]. In the case of this investigation, it was found that the IL-17 A values were generally higher in the CBG treatments, especially at 12 h; however, they did not turn out to be statistically different from the control.

In the case of IL-12p70 as an inflammation precursor cytokine, its production was between 15 and 20 pg/mL in response to both compounds, with a slight tendency to increase with time at all concentrations. There was no statistically significant difference. IL-12p70 production is essential for the induction of NK cells and the production of IFN-γ by Th1 lymphocytes. Relating to IFN-γ and IL-6, the production was between 20 and 35 pg/mL. There was no difference at any times or concentrations for CBD or CBG; this would corroborate that at small concentrations, or even 20 µM, the production of these pro-inflammatory cytokines was maintained, which turned out to be sustained over time, up to 12 h. There was only an overproduction of up to 45 pg/mL when CBD was used at 12 h of stimulation, while CBG at this same time and concentration was 32 pg/mL.

These two cytokines are generally produced by innate immune system cells as the first signal against different stimuli. Although there is no model in cells similar to PLF to be able to compare results, it is essential to note that most of the models in which the action of CBD has been sought have been models of inflammatory diseases such as diabetes [51], asthma [52], and pancreatitis [19], where it has been possible to show that circulating IL-6 and IFN-γ decrease. This can be attributed to the fact that they may be early physiological effects. However, there are animal models that allow prolonged systemic administration of the drug, which find that the pro-inflammatory markers are significantly reduced and the anti-inflammatory markers are increased.

In the current study, although a higher production of circulating IL-6 and IFN-γ was observed, it was only slightly higher than that of the control group. These values were maintained from the beginning, even with the minimum concentrations. Similarly, across the evaluated time periods, a trend towards similar behavior was observed, with no statistically significant differences noted between the compounds. It can be suggested that with the production of these two cytokines, a longer period of activation could be necessary to be able to show an eventual regulation, as stated in the studies mentioned above where the regulation of genes that code for the production of pro-inflammatory cytokines are seen with times of 24, 48, and 72 h [22,53,54].

One of the latest uses that has been tested for the CBD extract was developed in an in vitro model of lung epithelial cells infected by COVID-19, at a concentration of 9.5 to 10.9 µM, which managed to significantly reduce the production of IL- 6 and IL-8 without generating significant cytotoxicity, resulting in even better action than that of dexamethasone, which requires a dose greater than 12 µM [55].

Another hypothesis has also been generated in which the induction of the production of specific proinflammatory cytokines by CBD is related to the fact that, perhaps under some conditions, the increase in IFN-γ, which was also found in this investigation, would increase the genes that respond to IFN-γ and could also attenuate the proliferation of T cells. This theory was proposed by Kozela et al. in 2016, and from it, we can interpret that even though a proinflammatory cytokine was increased, its consequence may be immunosuppression [31]. Another example would be the production of IL-2, which, with a low stimulus, contributed to the appropriate environment to drive Treg induction, showing that the generation of pro-inflammatory cytokines may occur naturally in the reaction process against the extract but, in the correct doses and times, can induce immunosuppression [56].

Regarding the production of anti-inflammatory/regulatory cytokines such as IL-10 and IL-4, and TGF-β, which are crucial for healing, cell signaling, and repair processes, a notable discovery is that the production of IL-10 ranged between 15 and 20 pg/mL at concentrations of 0.5 µM after 3 and 6 h for both compounds assessed. Once the concentration was increased, its production tended to decrease to a maximum of 10 pg/mL. However, there was an increase in the production of IL-10 at a concentration of 10 µM after 12 h of stimulation for both compounds: between 15 and 20 pg/mL. This is an expected result since this cytokine is generally produced late, and the evaluation times were short compared with other studies.

The production of TGF-β was significantly higher when the stimulus was CBD, at a concentration of 0.5 µM at 3 h, while TGF-β production was significantly higher when stimulated by CBG at 12 h at concentrations of 0.5 µM, 10 µM, and 20 µM, compared with CBD. Although the exposure times in this research were shorter, these findings would agree with Rawal et al. in 2011, who found that low concentrations of CBD between 0.01 and 0.05 µM increased the production of TGF-β by up to 40%, and that production tends to decrease at concentrations greater than 4 µM to 30 µM. These values did not change significantly after 24 h [22]. In the study by Chiricosta et al., the production of TNF-β, mediated by the FURIN protein encoded by the FURIN gene which promotes the functional activation of TGF-β, an up-regulation of the gene was found after the treatment with CBD and moringa. However, this study found that, particularly for CBD, the highest production of TGF-β was related to a concentration of 3 µM, while these authors did so with 4 µM onwards [47].

IL-4 acts as an anti-inflammatory mediator by blocking the synthesis of IL-1, TNF-alpha, and IL-6. CBG concentrations of 10 µM and 20 µM, at all times evaluated, caused a significantly higher production of IL-4 by PLF (*p* < 0.05) compared with CBD. In the literature, this cytokine is not commonly evaluated. These findings may suggest the possibility of stimulating the production of IL-4 in this cell line with CBG stimulation at 10 µM and 20 µM. It can be associated with another immune regulation mechanism related to its action of blocking the synthesis of central proinflammatory cytokines.

Gingival fibroblasts and fibroblasts derived from the periodontal ligament are phenotypically different cells. PLF have a more homogeneous production of collagen and fibronectin in vitro (99% positive for collagen/fibronectin vs. 57% of gingival fibroblasts), added to the regenerative capacity of the periodontal ligament and its different gene expression [57]. Therefore, it is crucial to consider extending the research to other cell lines and also to analyze how they may behave differently in the tissues than in the culture media [58].

In terms of cytotoxicity, it was anticipated that there were not any differences for cell viability in the presence of CBD [22]. CBD showed low toxicity on PLF since, in concentrations between 0.5 and 20 µM, cell viability remained above 90% at the times evaluated, up to a maximum of 12 h (Figure 1). Although the highest concentration used in this research was 20 µM and the maximum time was 12 h, the results indicate that the treated cells have similar cell viability to cells not treated with CBD. This is in line with a study by Chiricosta et al. in 2019 [47], who used a concentration of 5 µM at more extended times of 24, 48, and 72 h and did not present effects or alterations in cell viability. In studies such as Rawal’s in 2012 [22], serial concentrations between 0.01 µM and 30 µM were used for 1 to 6 days, finding that no significant alterations were obtained up to a maximum concentration of 3 µM. The viability did not decrease by 90% in the following concentrations.

On the contrary, other authors stated that the populations of human oral cells, when exposed to CBD, have a significant reduction in viability from a concentration equal to or greater than 10 µM [54]. Our results suggest that for therapeutic doses of CBD, the concentrations should be, at least for this cell line, within this range of up to 20 µM. CBD has shown cytotoxic activity against different cancer cells, especially against breast cancer cells with IC_50_ values of 6 µM [59]. An important aspect of our analysis is that it was carried out using pure CBD extract; however, the Cannabis plant in its composition has terpenes that can significantly affect cytotoxicity in this cell line and whose mixture can change the results obtained [60].

The evidence for cytotoxicity regarding CBG is limited, as this is not a major component of the plant. In this case, the studies found are related to human cells associated with diseases. In peritoneal macrophages with a concentration of 1 µM for 24 h, CBG had no significant cytotoxic effect [61]. In motor neurons, concentrations between 2.5 and 7.5 µM showed a minimal decrease in cell viability [43]. This research presents cytotoxicity assays on oral cells as something new, particularly PLF, since there is no evidence in the current literature. Like CBD, CBG can be described as having low cytotoxicity.

When comparing CBD and CBG in this study, there was a statistically significant difference in concentrations of 3 µM at 12 h, with CBD having higher viability. Relating to cell viability after CBG treatment, a tendency to decrease up to 3 µM was evidenced; however, in concentrations greater than 10 µM and 20 µM, cell viability was higher. This may be associated with a rebound effect at these low doses or it is an aspect that should be corroborated with other techniques for measuring cell viability vs. cytotoxicity, such as succinimidyl-carboxyfluorescein ester (CFSE). In general terms, the two compounds have low cytotoxicity since, at higher concentrations, the mean viability values were higher than 85%. For the formulation of products for topical use, such as oral rinses, the average permitted cytotoxicity values would be greater than 70%. However, the current difficulty with oral rinse components is that those with an antibacterial effect turn out cytotoxic and vice versa [62,63]. These results show us that, at the concentrations and times evaluated, the organic compounds of CBD and CBG had acceptable levels of cytotoxicity. It is recommended to expand the concentrations, the exposure time, and the cells in which the evaluation would be made, given the complexity of the oral environment, in terms of the diversity of cell types that would encounter the product, such as keratinocytes and osteoblasts.

Concerning the mechanisms of action for CBD and CBG on PLF, we believe they could be linked to the recent discovery of CB2 receptors, as well as CB1 receptors’ expression, which are distributed throughout periodontal tissue, including gingival fibroblasts, connective tissue cells, and osteoblasts. CB2 receptors in periodontal tissue are distributed differently, as they are more abundant than CB1 receptors and are found predominantly in the junctional epithelium, in gingival connective tissue, in periodontal ligament fibroblasts, and on the alveolar bone surface. As extra information, it has been reported that with a THC stimulus (1 µM), it was possible to induce the migration of periodontal ligament cells towards a site with a previously created defect, evidencing cell migration and proliferation after 3 h of stimulation associated with the activation of the FAK pathway. In this way, it can regulate the MAP kinase pathway. A recently described receptor, GPR55, requires further investigation as it has not been investigated in periodontal tissue [64]. The presence of these two receptors, CB1 and CB2, in PLFs opens a door for analyses of different kinds of periodontal cells that could be sensitive to the action of CBD and CBG for immunological regulation in the periodontal environment.

## 5. Concluding Remarks

The minimal inhibitory concentrations of CBD and CBG against *S. mutans* were determined as 20 and 10 µM, respectively, whereas *C. albicans* was not affected by these compounds at concentrations up to 640 µM. Both substances decrease the metabolic activity of the subgingival multispecies biofilm, with CBD having a stronger inhibitory effect (50.38%). However, on pre-formed biofilms, both substances did not present any significant effect.

In terms of anti-inflammatory effect, proinflammatory cytokines (IL-1β, TNFα, IL-2, IL-17A, CCL2, CXCL10, CXCL8) production were not altered by any of the treatment conditions; however, CBD (10–20 µM) increased IFNγ release. On the other hand, production of the anti-inflammatory IL-10 cytokine was stimulated by CBD and CBG, while TGFβ and IL-4, two anti-inflammatory and regulatory cytokines, were increased by CBG. At last, both compounds presented no toxicity for fibroblasts.

The role of bacteria, together with the production of anti-inflammatory mediators and the maintenance of the viability of mammalian cells from the oral cavity, make these substances promising for clinical use and should be considered for future in vivo studies. In the near future, it will be useful to study Cannabis derivatives uses on biofilm formation as well as to functionalize different regeneration biomaterials with cannabinoids, which could be a useful approach to improve clinical outcomes after periodontal therapy.

## Figures and Tables

**Figure 1 antibiotics-13-00342-f001:**
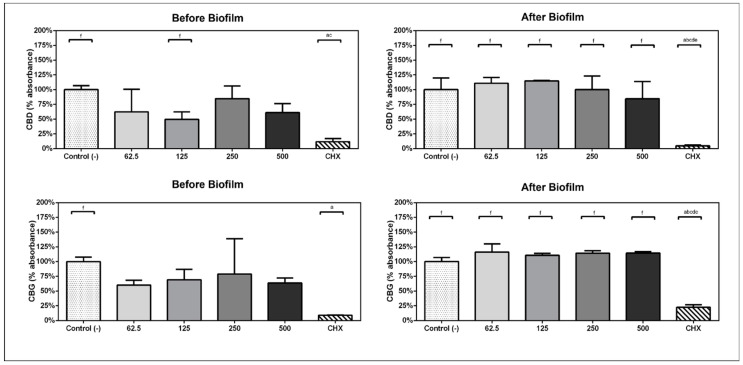
Comparison of percentage of absorbance with respect to the viability mean of the negative control group in experimental design A and B. Analysis performed with one-way ANOVA and Bonferroni post hoc. (a) statistically significant differences with c-treatment, (b) statistically significant differences with 125, (c) statistically significant differences with 250, (d) statistically significant differences with 500, (e) statistically significant differences with 62.5, (f) statistically significant differences with CHX.

**Figure 2 antibiotics-13-00342-f002:**
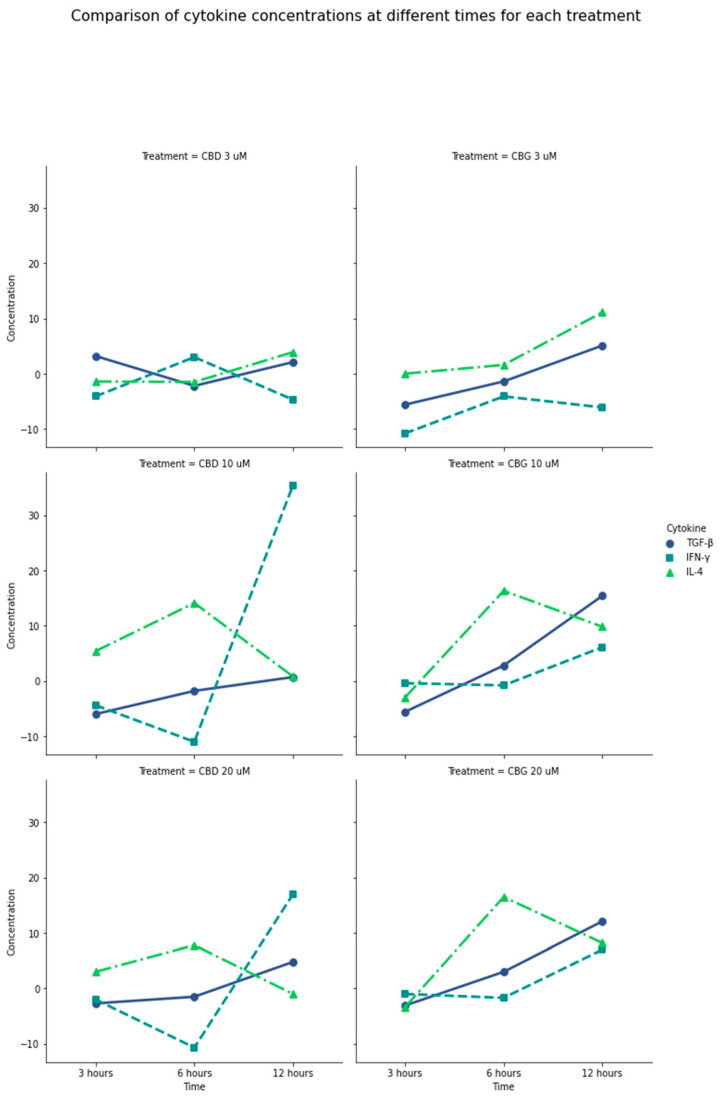
General comparison of IL-4, INF-γ, and TGF-β production in the presence of CBD and CBG. Difference between the amount of cytokine produced by PLF with CBD and CBG stimulation vs. the 0.01% DMSO control. Negative values on the plot mean that the cytokine was produced more with the control than with the treatment. CBG at 6 and 12 h at a concentration of 10 µM has a considerable production of anti-inflammatory mediators such as TGF-β and IL-4 and a controlled production of IFN-γ, compared with CBD at the same concentrations and times. CBD at 10 µM to 20 µM produced the highest amount of IFN-γ. TGF-β production was higher in the presence of CBG at the longest exposure time of 12 h and concentrations between 10 and 20 µM. It is possible to show that the production of cytokines is increased in concentrations higher than 10 µM; in lower concentrations, the production turns out to be low.

**Figure 3 antibiotics-13-00342-f003:**
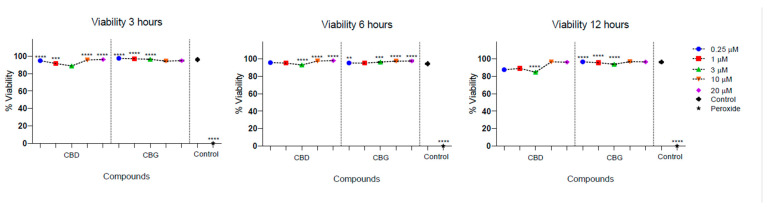
Cell viability of periodontal ligament fibroblasts in the presence of CBD and CBG. Cell viability on FLP cells was evaluated using CBG and CBD at concentrations of 0.25, 1, 3, 10, and 20 µm at 3, 6, and 12 h. The cell viability of the cells in the growth medium was used as a viability control. The cells were incubated with peroxide as a control for cell death. One-way ANOVA to determine the significant difference ** *p* < 0.01; *** *p* < 0.001; **** *p* < 0.0001.

**Table 1 antibiotics-13-00342-t001:** MIC values of CBD and CBG on planktonic cultures of *S. mutans* and *C. albicans.* CBD: Cannabidiol. CBG: Cannabigerol. CHX: Chlorhexidine. AMB: Amphotericin B. FLZ: Fluconazole. NA: Does not apply.

Compound		MIC
	*S. mutans*	*C. albicans*
CBD	20 µM	>640 µM
CBG	10 µM	>640 µM
CHX	4.68 µM	4.68–916 µM
AMB	NA	0.0625 µg/mL
FLZ	NA	0.5 µg/mL

## Data Availability

Data are contained within the article and Appendix A. Other data are unavailable due to privacity.

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
