# Peer review of "Antibiofilm and Immune-Modulatory Activity of Cannabidiol and Cannabigerol in Oral Environments—In Vitro Study"

_antibiotics, 2024, doi:10.3390/antibiotics13040342_

Round 1

Reviewer 1 Report

Comments and Suggestions for Authors

In Article “In vitro antibiofilm and immune-modulatory activity of cannabidiol and cannabigerol.” reported the SAR study reported that the CBD treated cell viability was greater than 95% and for CBG was higher than 88%. MIC for S. mutans with CBD was 20 µM, and 10 µM for CBG. For C. albicans, no inhibitory effect was observed. Multispecies biofilm metabolic activity Metabolic activity of multispecies biofilm was reduced by 50.38% with CBD at 125 µg/mL (p=0.03) and 39.9% with CBG at 62 µg/mL (p=0.023). CBD exposure at 500 µg/mL reduced the metabolic activity of the formed biofilm by 15.41%, but CBG didn´t have an effect. CBG at 6 and 12 hours at 10 µM has considerable production of anti-inflammatory mediators such as TGF-β and IL-4 at 12 hours. CBD at 10 µM to 20 µM produced the highest amount of IFN-γ. Conclusion: Both CBG and CBD inhibit S. mutans; they also moderately lower the metabolic activity of multispecies biofilms that are forming; however, CBD had a low effect on biofilms that had already developed. This result also provided role on bacteria, together with the production of anti-inflammatory mediators and the maintenance of the viability of mammalian cells from the oral cavity, make these substances promising for clinical use and should be considered for future studies.

Following minor correction are recommended:

(a)    The fonts of writing paper are not uniform and in some paragraphs the font is mixing of two or three font sizes because of copy paste error. The page number 1 and 2, line number 48 to 50 is different. Check the all-font size and correct it.

(b)   In reference section: The cited reference must be following journal guidelines. The year, volume and page number need to be corrected according to journal guidelines in all cited reference.

Author Response

We greatly appreciate the opportunity to improve the manuscript through corrections and comments made by evaluators. We have responded to all the reviewers' recommendations, and we have attached a new version with the changes incorporated, which are highlighted inside the document.

Reviewer #1:

  • The fonts of writing paper are not uniform and in some paragraphs the font is mixing of two or three font sizes because of copy paste error. The page number 1 and 2, line number 48 to 50 is different. Check the all-font size and correct it.

R/: We thank the evaluator. We make all the suggested editing modifications.

(b)   In reference section: The cited reference must be following journal guidelines. The year, volume and page number need to be corrected according to journal guidelines in all cited reference.

R/: We thank the evaluator. The references were updated and reviewed.

Reviewer 2 Report

Comments and Suggestions for Authors

Dear author

Is a good manuscript, just there are some point to improve.

In abstract, include abbreviation of cannanidiol (CBD), cannabigerol (CBG), please

Line 29. Metabolic change by metabolic

Line 46. Cannabis Sativa change by Cannabis sativa

Line 48-50. Formato of text

Line 90. Should amphotericin be capitalized?

Line 216. Expand this method, for example mention dosaje

Line 269. Improve the way you present your results (subtitle 3.4)

Line 306. Check, subtitle 3.5 is moved

Line 322. Maybe you should present this variable in a table

Line 348-351. Check, format please

Line 403-404. Edit paragraph

Line 542. Change Kit by kit

*You should read your manuscript and change the word regarding, is very used in all the text

*Improve your title, introduction and conclusions

*Improve introduction to give importance to biofilm that can lead to problems such as inflammation (expand the inflammation part)

Regards

Author Response

We greatly appreciate the opportunity to improve the manuscript through corrections and comments made by evaluators. We have responded to all the reviewers' recommendations, and we have attached a new version with the changes incorporated, which are highlighted inside the document.

Reviewer #2:

1.In abstract, include abbreviation of cannanidiol (CBD), cannabigerol (CBG), please

R/: We thank the evaluator. The correction was made and can be verified in the new version of the manuscript.

  1. Line 29. Metabolic change by metabolic

R/: We thank the evaluator. The correction was made and can be verified in the new version of the manuscript

  1. Line 46. Cannabis Sativa change by Cannabis sativa

R/: We thank the evaluator. The correction was made and can be verified in the new version of the manuscript

  1. Line 48-50. Formato of text

R/: We thank the evaluator. We make the suggested modifications.

  1. Line 90. Should amphotericin be capitalized?

R/: We thank the evaluator. It was reviewed in MeSH and Pubmed, and the usual form is  lower case.

  1. Line 216. Expand this method, for example mention dosage

R/: We thank the evaluator. The paragraph was rewritten as the information on the technique was duplicated.

  1. Line 269. Improve the way you present your results (subtitle 3.4)

R/: Thanks for the suggestion to improve our manuscript. The 3.4 section was rewritten. You can find the changes highlighted in the manuscript.

  1. Line 306. Check, subtitle 3.5 is moved

R/: The correction was made and can be verified in the new version of the manuscript.

  1. Line 322. Maybe you should present this variable in a table

R/: Thank you very much for the suggestion. We will add the table, but we kindly ask you to reconsider this suggestion as we think the figure easy reflect the results. We suggest to add the new table as supplementary material, but it could be redundant information. The new table is Table 2.

  1. Line 348-351. Check, format please

R/: The correction was made and can be verified in the new version of the manuscript.

  1. Line 403-404. Edit paragraph

R/: We thank you for the suggestion. The paragraph was edited

  1. Line 542. Change Kit by kit

R/: The correction was made and can be verified in the new version of the manuscript.

  1. *You should read your manuscript and change the word regarding, is very used in all the text

R/: We thank the reviewer. The manuscript was revised and paraphrased.

  1. *Improve your title, introduction and conclusions

R/: We thank the evaluator. The correction was made and can be verified in the new version of the manuscript.

New title: Antibiofilm and immune-modulatory activity of cannabidiol and cannabigerol in oral environments. In vitro study

Introduction: new paragraphs were added and the sequence was changed.

Concluding remarks were modified as requested.

  1. *Improve introduction to give importance to biofilm that can lead to problems such as inflammation (expand the inflammation part)

R/: We thank the evaluator. The correction was made and can be verified in the new version of the manuscript. New paragraphs have been included:

Therapeutic approaches with substances with concomitant antimicrobial and   anti-inflammatory properties are widely appreciated today since the spectrum of inflammatory diseases initiated by infectious components is increasingly broader and dysbiotic processes are becoming more clearly associated with inflammatory conditions every day (6, 7), including oral diseases such as periodontitis (8, 9). The above, together with the great concern worldwide about the excessive use of antibiotics that has led to unprecedented levels of bacterial resistance (10, 11), highlights the need to address these pathologies from both arms of the problem, inflammation and antimicrobial strategies without the harming secondary effects of the antibiotics.

Additionally, there is the fact of the difficulty in treating biofilm-related diseases (12, 13) (one of the main causes of deaths at the hospital level), given the characteristics of these complex structures that mean that the bacterial communities embedded in polymeric matrices are protected. against the action of antimicrobials (14, 15). Bearing this in mind, there is a very important field of action for the use of cannabis derivatives in the treatment of infectious/inflammatory diseases.

Reviewer 3 Report

Comments and Suggestions for Authors

Is the title too broad? Maybe make it more specific, like focusing on treating oral pathologies.

In the introduction, briefly discuss the current limitations of using conventional antibiotics for treating oral pathologies. This would make the case stronger for looking into alternative therapies like CBD and CBG.

In the discussion, the manuscript mentions that CBD and CBG moderately decrease the metabolic activity of multispecies biofilms. However, it would be helpful to talk about the real-world significance of this finding. Can this moderate effect actually lead to a noticeable reduction in plaque formation in vivo?

The manuscript acknowledges the need for further research on how CBD and CBG affect the structural cells that protect and support periodontal tissues. It would be good to suggest specific directions for future studies in the concluding remarks. For example, upcoming in vivo studies could look into the effects of applying topical CBD or CBG on periodontal tissue health.

Comments on the Quality of English Language

It looks like the sentence in lines 29-31 is incomplete.

In lines 32 and 33, was it 12 hours or 6 and 12 hours?

I would suggest removing the full names of CBD and CBG in lines 80-81 since these abbreviations were already introduced earlier.

Regarding the Calgary Biofilm Device (CBD) mentioned in line 171, I would avoid using the same abbreviation (CBD) for different terms to prevent confusion.

The title of section 3.5, "Cytotoxicity Activity" (line 306), should be moved to a new paragraph by itself.

Please fix the typo "C.BG" (line 323).

Author Response

We greatly appreciate the opportunity to improve the manuscript through corrections and comments made by evaluators. We have responded to all the reviewers' recommendations, and we have attached a new version with the changes incorporated, which are highlighted inside the document.

Reviewer #3:

1.Is the title too broad? Maybe make it more specific, like focusing on treating oral pathologies.

R/: We thank the evaluator. The title was adapted and can be verified in the new version of the manuscript

  1. In the introduction, briefly discuss the current limitations of using conventional antibiotics for treating oral pathologies. This would make the case stronger for looking into alternative therapies like CBD and CBG.

R/: We thank the evaluator. The correction was made and can be verified in the new version of the manuscript. New paragraphs were added.

  1. In the discussion, the manuscript mentions that CBD and CBG moderately decrease the metabolic activity of multispecies biofilms. However, it would be helpful to talk about the real-world significance of this finding. Can this moderate effect actually lead to a noticeable reduction in plaque formation in vivo?

R/: We thank the evaluator. We include the following text in discussion:

“Future studies should evaluate whether this moderate antibiofilm outcome would be effective in vivo. Typically, the first step in natural product bioprospecting is in vitro assays. Subsequently, the natural product-derived molecules should undergo in vivo studies, including animal models, and ultimately clinical trials. Vehicle, route of administration and pharmacokinetic properties can all influence whether the positive effects observed in vitro will be reflected in vivo”

  1. The manuscript acknowledges the need for further research on how CBD and CBG affect the structural cells that protect and support periodontal tissues. It would be good to suggest specific directions for future studies in the concluding remarks. For example, upcoming in vivo studies could look into the effects of applying topical CBD or CBG on periodontal tissue health.

R/: Thank you. Following your suggestion, we included new information on future perspectives.

Comments on the Quality of English Language:

  1. It looks like the sentence in lines 29-31 is incomplete.

R/: We thank the evaluator. we revised the English.

  1. In lines 32 and 33, was it 12 hours or 6 and 12 hours?

R/: We thank the evaluator. The correction was made and can be verified in the new version of the manuscript

  1. I would suggest removing the full names of CBD and CBG in lines 80-81 since these abbreviations were already introduced earlier.

R/: We thank the evaluator. The correction was made and can be verified in the new version of the manuscript

  1. Regarding the Calgary Biofilm Device (CBD) mentioned in line 171, I would avoid using the same abbreviation (CBD) for different terms to prevent confusion.

R/: We thank the evaluator. The correction was made and can be verified in the new version of the manuscript

  1. The title of section 3.5, "Cytotoxicity Activity" (line 306), should be moved to a new paragraph by itself.

R/: We thank the evaluator. The correction was made and can be verified in the new version of the manuscript

10.Please fix the typo "C.BG" (line 323).

R/: We thank the evaluator. The correction was made and can be verified in the new version of the manuscript